# Environmental Selection Shapes Bacterial Community Composition in Traditionally Fermented Maize-Based Foods from Benin, Tanzania and Zambia

**DOI:** 10.3390/microorganisms10071354

**Published:** 2022-07-05

**Authors:** Maarten de Jong, Anna Y. Alekseeva, Kulwa F. Miraji, Sydney Phiri, Anita R. Linnemann, Sijmen E. Schoustra

**Affiliations:** 1Laboratory of Genetics, Wageningen University and Research, P.O. Box 17, 6700 AA Wageningen, The Netherlands; ir.m.n.dejong@gmail.com (M.d.J.); anna.alekseeva.msu@gmail.com (A.Y.A.); 2Tanzania Agricultural Research Institute, Ifakara Centre, Ifakara, Tanzania; kfurahisha@gmail.com; 3Department of Food Science and Nutrition, School of Agricultural Sciences, University of Zambia, Lusaka 10101, Zambia; nysydph@gmail.com; 4Food Quality and Design, Wageningen University and Research, P.O. Box 17, 6700 AA Wageningen, The Netherlands; anita.linnemann@wur.nl

**Keywords:** Munkoyo, Mawè, Aklui, Togwa, traditional fermentation, species sorting, 16S rDNA amplicon sequencing

## Abstract

Natural (microbial) communities are complex ecosystems with many interactions and cross-dependencies. Among other factors, selection pressures from the environment are thought to drive the composition and functionality of microbial communities. Fermented foods, when processed using non-industrial methods, harbor such natural microbial communities. In non-alcoholic fermented foods the fermenting microbiota is commonly dominated by 4–10 species of bacteria, which make them suitable model systems to study ecosystem assembly and functioning. In this study, we assess the influence of the environment on the composition of microbial communities of traditional fermented products from Africa. We compare differences between microbial communities that are found in similar products but come from different countries, hypothesizing they experience different environmental selection pressures. We analyzed bacterial community composition in 36 samples of various cereal-based fermented foods from Benin, Tanzania and Zambia using 16S rDNA amplicon sequencing. The differential abundance analysis indicates that the bacterial communities of fermented foods from the three countries are dominated by mostly lactic acid bacteria belonging to the genera of *Lactobacillus*, *Weisella* and *Curvibacter*. The samples from Zambia contain the most dissimilar microbial communities in comparison with samples from Benin and Tanzania. We propose this is caused by the relatively low temperature in Zambia, suggesting that indeed environmental selection can shape community composition of fermenting microbes.

## 1. Introduction

Microbial communities as ecosystems are widespread and diverse: they are represented by, e.g., stromatolites, soil, microbial communities of the gut or skin in humans/animals and contain a variety of different species. Selection pressures from the environment have been suggested to influence the community structure of these ecosystems in a process called species sorting [1]. However, systematic studies on what types of selection pressures are most relevant for specific ecosystems are limited in number [2]. The reason is that natural communities are too complex, and many factors affect their composition. Recent studies showed that traditional fermented foods could be used as a suitable model system to study the dynamics of community composition and functionality of microbial ecosystems: such communities are diverse, but still manageable in terms of tracking their composition, as well as factors which affect their composition [1,2].

During food fermentation, micro-organisms such as bacteria or yeast convert raw materials into food products with increased nutritional value, prolonged shelf-life and enhanced sensory properties. While food fermentation is a very old practice [3], many well-known fermented foods such as yoghurt and beer are now manufactured under highly controlled (industrial) conditions using a defined starter cultures containing few microbial species [4]. In contrast, fermented foods contain microbial communities of high microbial species diversity [5]. This rich microbial diversity contribute to probiotic properties of the fermented foods. In addition, these traditional foods are highly resistant to spoilage [6].

Despite the lack of controlled conditions, the composition of microbial communities in the traditionally fermented foods is stable; the microbial community composition eventually reaches a steady state from batch to batch. This is caused by the process of back-slopping: the addition of starter culture by transferring some of an old batch to fresh raw material. Co-adapted, evolutionarily fit microbes from the previous cycle dominate the new batch, and repeated cycles result in adapted, evolutionarily stable communities. In the absence of sample exchange between producers, microbial ecosystems from the different producers evolve independently [6]. Previous work has shown that initial differences between microbial community structure at the start of a fermentation cycle does not significantly impact the final community structure [7], and that processing practice is an important driver of microbial community composition [8]. Work based on laboratory experiments has suggested that processing temperature is a key factor, which can drive microbial community composition [9]. This implies that the environment, where fermentation takes place, may be important in shaping microbial community structure—this idea has inspired the present work.

To address a question on the role of the environment in shaping bacterial community structure, we compared bacterial communities of samples of similar foods from different countries. In this study, a total of 36 samples of five cereal-based traditionally fermented products were bought from local markets in Benin, Tanzania and Zambia. The products were made according to the traditional steps of mixing the cereal with water, boiling the mixture (except in the case of the fermented dough mawè, which is not boiled) and subsequent fermentation assisted by plant-derived enzymes for around 48 h. Dissimilarities among the samples included different geographic locations and seasons, and different amylase enzyme sources. For some of the samples from Zambia, the amylases came from Rhynchosia roots, as opposed to the products from Benin and Tanzania where malted cereal or sweet potato peel was used. We hypothesized that differences between the countries would translate to different selection pressures, and that we would find dissimilar microbial communities in the samples from the different countries.

To profile the microbial communities in the product samples, DNA was extracted and regions of the marker-gene 16S rDNA were amplified and sequenced. The amplicon sequences corresponding to the different samples formed the basis of the bioinformatics analysis. The novel, and one of the most accurate bioinformatics pipelines for amplicon sequencing data, QIIME2 was used to perform major data-processing steps [10,11]. The amplicon sequences were dereplicated and transformed into amplicon sequence variants (ASVs) and later mapped to bacterial genera. These were used as input for R-packages that estimated alpha diversities, beta diversities between the samples and differential abundances of ASVs and genera in samples across countries of origin.

## 2. Materials and Methods

### 2.1. Sampling, DNA Extraction and Sequencing

Traditionally fermented foods were bought from local markets in Benin, Tanzania and Zambia. All foods were produced at non-industrial scale and without the use of defined starter cultures. Twelve samples from each country accounted for 36 samples total. The foods bought in Tanzania were Togwa, and in Zambia the foods were Munkoyo. The samples from Benin were more diverse, comprising three Aklui samples, four Mawè samples and five Akpan samples. All products were cereal-based, either maize or millet (full metadata in Appendix A). DNA was extracted following previously designed protocol [6]. One mL of sample was centrifuged at high speed. Next, 500 µL of TESL (TESL is (25 mM Tris, 10 mM EDTA, 20% sucrose, 20 mg/mL lysozyme), 10 μL mutanolysin and 100 μL lysozyme were added to the pellet, followed by incubation at 36 °C and light shaking. Next, 500 μL of GES reagent was added (GES is 5 M guanidium thiocyanate, 100 mM EDTA, 0.5% sarkosyl), and the mixture was cooled on ice for five minutes. After that, cold ammonium acetate was added and the mixture was kept on ice for 10 min. The mixture was then centrifuged, the supernatant was collected and further purified by adding chloroform-2-pentanol in a 1:1 ratio, and spinning at 12,000 RPM. The DNA in supernatant was precipitated by adding 0.1 Volume of 3 M sodium acetate and 2.5 Volume of 100% ethanol and storing the mixture overnight at 20 °C. Next, the mixture was spun for 20 min at 12,000 RPM at 4 °C. The supernatant was removed and the DNA pellet was washed out using 1 mL of cold 70% ethanol. The sample was then spun for 10 more minutes at 12,000 RPM at 4 °C and the resulting supernatant was again removed. Finally, the DNA pellet was air dried for 10 min at room temperature and dissolved in 10 mM tris pH 7.5. DNA QC, library preparation and paired-end Illumina MiSeq sequencing was performed by LC Sciences. Universal primers 338F and 806R designed to target the V3—V4 region of the 16S rDNA generated an amplicon length of about 469 basepairs (bp) in length. LC Sciences provided the paired-end sequence reads in non-multiplexed format, stored in two FASTQ files per sample.

### 2.2. Data Processing

The paired-end sequence reads were processed using plugins of ‘Quantitative Insights Into Microbial Ecology’ 2: QIIME2 version 2019.4 [10]. This package was installed on a local server into a conda environment supported by Miniconda3. It supports two mature interfaces: a Linux command line interface and a Python 3 Application Programmer Interface (API), referred to as the Artifact API. The latter was used in this project. Here, ‘Artifacts’ correspond to QIIME2 Python objects that contain data and metadata (in the command-line environment, they correspond to compressed files with data and metadata). QIIME2 was used for de-noising, dereplication, taxonomic classification and phylogenetic inference.

#### 2.2.1. Sequence-Quality, Denoising and Dereplication

The overall sequence quality information in the FASTQ-files was visualized using demux version 1.24.0, a QIIME2 plugin that also supports demultiplexing. The demux-visualizer ‘summarize’ was used to build a visualization-Artifact with plots of sequence quality (Appendix A). The plots were used as a visual guide in the succeeding step: de-noising the sequences. De-noising was performed using ‘Divisive Amplicon Denoising Algorithm’ 2: DADA2 version 2019.4.0 [12]. DADA2 performed truncation, trimming, chimera removal and dereplication. The sequence truncation conserved the first 290 bp of the forward reads and the first 205 bp of the reverse reads, informed by the sequence quality plots.

The first 20 bp were trimmed off for both forward and reverse reads. Other DADA2 parameters were left to the default. Dereplication constitutes the merging of identical or highly similar sequences into single ‘features’, each with a distinctive feature ID, representative DNA sequence and a number of corresponding sequences (frequency or count). No further clustering was performed.

#### 2.2.2. Taxonomic Classification

The taxonomies of the ASVs were estimated using a Naïve Bayes classification model. Training data was downloaded from the Ribosomal Database Project (RDP), version 16 [13]. Three methods from the plugin ‘feature-classifier’ version 2019.4.0 [14] were used for the taxonomic classification: extract-reads, fit-classifier-naïve-bayes and fit-classifier-sklearn [15]. Extract-reads were used to extract read-like fragments from the RDP reference sequences, based on the forward and reverse primers. These read-like sequence fragments and the RDP reference taxonomies represented the independent and dependent variables used to fit the Naïve Bayes classifier. The classifier was fit to the read-like sequence fragments and the corresponding RDP taxonomies using fit-naïve-bayes function. By providing the representative sequences of the ASVs to the classifier, the taxonomic classifications of the ASVs were obtained with a minimal confidence of 70% (below which, taxonomy was declared undetermined).

#### 2.2.3. Phylogenetic Inference

Phylogenetic relationships of the representative ASV sequences were inferred in two steps: multiple sequence alignment (MSA) and phylogenetic tree construction. MSA was performed using ‘Multiple sequence Alignment using Fast Fourier Transform’: MAFFT version 7 [16]. Positional conservation and gap filtering of the MSA, also known as ‘masking’, was performed using Mask with the default parameters: maximum gap frequency 1.0 and minimum conservation 0.4 [17]. The masked MSA was used as input for bootstrap supported phylogenetic tree construction with IQTree-Ultrafast Bootstrap version 2019.4.0 [18,19]. The specified random seed value (applied for reproducibility) was 42. The nucleotide substitution model was determined automatically with Modelfinder and the number of generated bootstrap replicates was 1000 [20]. Splits with bootstrap supports below 50% were removed from the phylogenetic tree by the merging of those splits. This was performed in R 3.6.1 using collapseUnsupportedEdges from the R-package ‘ips’, short for ‘interfaces to phylogenetic software’, version 0.0.11 [21].

### 2.3. Statistical Analysis

The acquired ASVs, taxonomic assignments and phylogenetic inferences were used for the statistical analysis of the sample microbiomes. All the analysis was performed in R version 3.6.1. The primary packages used for the analysis were phyloseq version 1.28.0 [22], vegan version 2.5–6 [23] and DESeq2 version 1.24.0 [24]. For most of the analysis, the community data was assigned to a phyloseq-object dubbed, which stored the data in four categories: the ASV-table; the sample data (metadata); the taxonomy table; and the representative DNA sequences.

### 2.4. Rarefaction

Rarefaction curves were attained using the function ‘rarecurve’ from phyloseq; ‘rarecurve’ was provided with the transpose of the ASV-table from the community data and step size for sample sizes was set to 50. The community data was rarefied using ‘rarefy_even_depths’ from phyloseq. The random seed value was set to 123, sample size to 90% of the smallest sample size, and random sampling was performed without replacement.

### 2.5. Alpha and Beta Diversities

The alpha diversities of the microbial community samples were visualized using the ‘plot_richness’ function of the phyloseq package. The phyloseq object was submitted as ‘physeq’, country was specified as variable to map the horizontal axis ‘x’, and two alpha diversity measures were specified under ‘measures’: Richness; corresponding to the number of unique ASVs present in a sample and Shannon; corresponding to the ‘Shannon diversity index’. The Shannon diversities incorporated the relative proportions of the distinct ASVs according to Equation (1), where s was the number of ASVs present in a sample and *p_i_* was the proportion of amplicons that belonged to the *i*th ASV of the sample.
(1)$$H^{\prime }=-\overset{s}{\underset{i=1}{\sum }}p_{i}\;ln\left(p_{i}\right)$$

The Shannon diversity index favors a high richness, and proportional abundance equality [25]. In order to visualize the datapoints in boxplots, ‘geom_boxplot’ from the ggplot2 R-package was added to the plot command. Differences between country diversities were investigated using Mann–Whitney U testing. The wilcox.test function in the stats package (version 3.6.1) was used in an unpaired setting and with approximate *p*-value computation.

The beta diversity among the 35 microbial community samples was visualized by calculating pairwise distances between samples, and by visualization of the multidimensional results using dimension reduction (ordination) and dendrogram visualization of inferred sample clusters. The distances between the samples were computed using the distance function from the Phyloseq package on the phyloseq object. The distance method specified was Unifrac [26]. The ordination was performed using the ordinate function from the Phyloseq package on the phyloseq object. The ordination method specified was Non-parametric Multidimensional Distance Scaling, or NMDS [27]. Hierarchical complete linkage clustering was performed on the Unifrac distance matrix [28], using ‘hclust’ from the ‘stats’ R-package version 3.6.1.

### 2.6. PERMANOVA

Permutational analysis of variance, PERMANOVA, [29], was performed using the ‘adonis’ function from the vegan R package [23]. The Unifrac dissimilarity matrix was specified as the dependent variable, and the countries as the independent, categorical variable (Equation (2)).
(2)Unifrac dissimilarities ~ Country

For the three PERMANOVA’s each comparing two countries, separate phyloseq objects with only the two countries were obtained using ‘subset_samples’ from the phyloseq package. The Unifrac distances were calculated for the subsampled phyloseq objects and the PERMANOVA’s were specified the same way as previously. Multiple testing correction was performed by adjusting the alpha value according to Bonferroni correction (Equation (3)) [30].
(3)$$\alpha_{Corrected}=\frac{\alpha }{N_{tests}}$$

### 2.7. Differential Abundancy Analysis

Venn-diagrams representing the overlapping and unique ASVs and genera were constructed using ‘venn.diagram’ from the R-package VennDiagram version 1.6.20. Differential abundancy analysis was performed using the R-package DESeq2 [24]. A few steps were performed to make the community data stored in the phyloseq object compatible with DESeq2. First, the phyloseq object was transformed using ‘transform_sample_counts’ from the phyloseq package, in order to add pseudo-counts of one to each cell in the ASV-table (this was required to prevent division-by-zero errors in the DESeq analysis). The conversion from phyloseq object to DESeq object was performed using the function ‘phyloseq_to_deseq2’, with the phyloseq object as data and ‘Country’ specified as independent variable. Next, the differential abundance analysis was run on the community data using the function DESeq from the DESeq2 package. Results were extracted for all three country combinations using the ‘results’ function from DESeq2, and an alpha value of 0.01 was used, applicable to the Benjamini–Hochberg adjusted *p*-values [31]. Using cbind, the taxonomic classifications of the phyloseq object were attached to the differential abundance object. The above was performed a second time, but this time on the 480 most abundant ASVs (accounting for 98% of the total abundancy), and the ASVs agglomerated to genera using the ‘tax_glom’ function from the phyloseq package.

## 3. Results

### 3.1. Amplicon Sequence Variants (ASVs)

The 35 samples with a total of 597,755 amplicon sequences were analyzed and merged into 1127 Amplicon Sequence Variants (ASVs) by DADA2 [12]. Table 1 shows ASV distribution. The mean ASV frequency (number of amplicons corresponding to a single ASV) was more than fifteen times as high as the median ASV frequency, and the most frequent ASV accounted for slightly over twelve percent of the total abundance. These numbers emphasized that most ASVs had a low frequency, and a small minority of ASVs accounted for most of the total abundance.

### 3.2. Rarefaction

To compare the samples based on their differential ASV abundances, the samples were rarified, namely, normalized to a fixed number of amplicons. After rarefaction, all samples corresponded to 4518 amplicon observations, selected randomly with replacement from the original, larger samples. A side-effect of the rarefaction was the removal of 57 low-abundance ASVs, as those were no longer present after subsampling. The rarefaction left a total of 1061 ASVs and 158,130 amplicon observations in the rarified ASV-table (Table 2).

The rarefaction curves (a.k.a. taxon sampling curves) of the samples were plotted before and after rarefaction to visualize the relationships between the numbers of amplicons and the numbers of ASVs in the samples. These relationships were assumed to be linear if only small fractions of the true number of ASVs were discovered, and asymptotic if most ASVs present in the samples were discovered, as novel ASV discoveries would become more rare [32]. The rarefaction curves of both the original samples and the smaller rarified samples followed asymptotic growth (Figure 1), indicating that the samples represented the microbial diversities adequately.

### 3.3. Community Composition

The 1061 ASVs in the 35 samples were classified into 182 different genera. Analogous to the ASVs frequency distribution; a small number of abundant genera accounted for a large portion of the total observations. Whereas half of the genera corresponded to 20 observations or less, the mean genus abundance was more than 40 times this with ~838 observations (Table 3).

The most abundant genus was Lactobacillus, with more than 35,000 observations, or about 23% of the total abundancy. The second most abundant genus was *Weisella*, adding another 16%, followed by *Curvibacter*. The top-three genera alone accounted for more than half of all the observations, and the fifteen most abundant genera account for almost 88%. The list of most abundant genera was composed of lactic acid bacteria such as *Lactococcus*, *Weisella*, *Streptococcus* and *Leuconostoc*, as well as proteobacteria such as *Curvibacter* and *Acetobacter* (Figure 2).

Phylogenetic relationships between bacterial groups in studied samples were estimated. The analysis was conducted in two steps: multiple sequence alignment (MSA) and phylogenetic tree construction. A phylogenetic tree based on amplicon sequences combined with SATé-enabled phylogenetic placement (SEPP) technique [14,19] allowed for the insertion of amplicon reads into a robust phylogenetic tree. For consistency, a phylogenetic tree using the RDP 16s rDNA training sequences and based on 92% abundance genera was created. The data shown on Appendix A.

### 3.4. Alpha and Beta Diversity in Microbial Communities

Alpha diversity corresponds to the biodiversity within a sample. Here, we applied two methods to represent alpha diversity: richness and Shannon diversity. Richness, or ‘Observed’, corresponded to the number of distinct ASVs present in the samples. The Shannon diversities incorporated the relative proportions of the distinct ASVs, favoring a high richness, and proportional abundance equality [25]. The richness and Shannon diversities were calculated and visualized in boxplots for the three different countries (Figure 3). The richness indicated a significantly (Mann–Whitney U-test) higher diversity for samples from Benin compared to samples from Tanzania (*p* = 0.01) and samples from Zambia (*p* = 0.001), while samples from Tanzania and Zambia were not significantly different. All richness values for Tanzania were within the range of the richness values of Zambia. The Shannon diversity revealed less variation between the plots, and no significant differences in diversities between countries (*p*-values based on these and more diversity measures in Appendix A).

Beta diversity refers to the difference in diversity between different samples. The beta diversity among the 35 microbial community samples was inferred from computed Unifrac distances [26]. These sample distances were visualized using NMDS ordination [27] and a hierarchical clustering dendrogram. In the NMDS ordination (Figure 4), the samples from Zambia were generally separated from the samples from Benin and Tanzania along the horizontal axis, while samples from Benin and Tanzania were generally separated along the vertical axis.

A complete linkage hierarchical clustering dendrogram revealed similar patterns (Figure 5), being defined by five large clusters. Most samples from Benin were found in the same cluster (the branch on the right), with the exception of B7 and B11. Most samples from Zambia were also located in the same cluster (third major branch from the left), with the exception of four samples that clustered with Tanzanian samples. Samples from Tanzania were the most scattered across the dendrogram. A contrasting feature of the dendrogram compared to the NMDS ordination plot, was the close proximity between Zambian samples Z3 and Z11 and Tanzanian samples T7 and T8 (left of the dendrogram). In the NMDS ordination plot, these pairs were stretched across the plane.

Of the 11 pairs of samples at the tips of the dendrogram, ten consisted of samples from the same country. The one exception was between the samples B10 and T5 (Figure 5).

### 3.5. PERMANOVA Differential Abundance Testing

In order to quantify the significance of the variation within and between countries, PERMutational ANalysis Of Variance, PERMANOVA [29], was performed on the microbial community data. Four PERMANOVA’s were performed, one incorporating samples from all the countries, and three comparing pairs of countries to each other. The following zero hypotheses and alternative hypotheses applied to the PERMANOVA’s (α = 0.05):

**Hypothesis** **0** **(H0).**
*The centroids and the dispersions (as defined by Unifrac dissimilarities) are equivalent for the countries.*


**Hypothesis** **a** **(Ha).**
*The centroids, or the dispersion, or both, are not equivalent between the countries.*


The first PERMANOVA incorporating all three countries revealed that ~18.5% of the total variation resided between the different countries (Table 4). The remaining variation (~81.5%) corresponded to the sum of the variation within the three countries individually. The resulting F-value (and degrees of freedom) resulted in a low probability value of 0.001 << α, leading to rejection of the zero hypothesis. The two PERMANOVA’S comparing samples from Zambia to samples from Benin and Tanzania resulted in similar statistics as the first PERMANOVA, whereas the PERMANOVA comparing samples from Benin to samples from Tanzania revealed a lower fraction of between-group variation, and as a result a lower F-value and higher probability value. Bonferroni multiple testing correction for the three pairwise tests [30], resulted in *α_Corrected_* ≅ 0.017. This threshold implied significant probability values and rejection of H0 for all three comparisons, but the H0 probability in the comparison between Benin and Tanzania samples was far higher than for the other two comparisons. These results revealed that the (centroids or dispersion or both of the) bacterial communities were not equivalent between the countries, and that samples from Zambia deviated most from the samples from the other two countries (Appendix A).

### 3.6. Differential ASV Abundance Analysis

The significant difference between bacterial communities from different countries should be reflected in the presence and abundance of ASVs and genera in the samples. A Venn diagram of the ASVs revealed that the great majority; about 84%, was unique to the respective country of origin (Figure 6). The number of genera unique to the respective country of origin accounted for a more modest proportion of the total number of genera, about 54% (Figure 6).

In order to assess how many of the 887 unique ASVs were significantly differentially abundant (DA), a differential abundance analysis was performed (Table 5). With the use of DESeq2 [24], the significant DA ASVs between different pairs of countries were identified. The analysis revealed that samples from Benin and Tanzania had 52 DA ASVs, whereas the comparisons between Benin and Zambia resulted in 91 DA ASVs, and the comparisons between Tanzania and Zambia resulted in 80 DA ASVs (Table 3). Most of the DA ASVs in the two comparisons with Zambia resulted from a low abundance or absence of the respective ASVs in Zambia.

Comparing Benin with Tanzania revealed both less DA ASVs, and more balanced distributions of respective overabundance (a 3:2 ratio as opposed to a 2:1 ratio found in the comparisons involving Zambia).

A more restrained DESeq2 analysis, in which only the 480 most abundant ASVs (accounting for 98% of the total abundance) were incorporated, resulted in three distinct genera overrepresented in the samples from Zambia: *Aeromonas*, *Enterobacter* and *Raoultella*. These genera are all part of the class Gammaproteobacteria, and *Enterobacter* and *Raoultella* share the same family: the Enterobacteriales. Bacteria from all three genera are classified as mesophiles; thriving in moderate temperatures [33,34], and 22 genera were underabundant in samples from Zambia compared to samples from Benin, Tanzania or both. Most of these genera corresponded to mesophilic bacteria.

Consequently, the underabundance in Zambia could not be explained by classifications of the optimal growth temperatures alone. There were three exceptions to this. *Acetobacter* and *Aquabacterium* genera classified as thermotolerant and thermophilic, respectively [35,36,37], were found to be overabundant in samples from Benin, compared to samples from both Tanzania and Zambia. Another thermophilic genus *Ralstonia*, was similarly overabundant in Benin compared to Tanzania and Zambia, but also in Tanzania compared to Zambia. *Ralstonia* was the most striking example of relative abundance of thermophilic or thermotolerant genera being most abundant in the warmest environments and vice versa. All three thermophiles appeared to be favored under higher temperature conditions present in the samples from Benin, and to a lesser extent Tanzania (Figure 7).

## 4. Discussion

The main hypothesis to motivate our study was that different environments in different countries would shape the bacterial community composition in otherwise comparable cereal-based traditional fermented foods processed in these countries. Our results, based on bacterial community profiles of 36 samples from three countries, provide support for this hypothesis. The traditional fermented foods from Zambia proved to harbor a significantly different microbial community than those from Benin and Tanzania (Appendix A).

There were two environmental or processing variables unique to the samples from Zambia: temperature and enzyme source. As shown in the results, the temperature of 16 °C at the time of sampling in Zambia was much lower than the temperatures at the times of sampling in Benin and Tanzania (29 and 25 °C, respectively). It could be hypothesized that the differences in bacterial communities between countries found in this study were caused by the presence of plant root material in some of the Zambian samples. The roots would bring soil-based bacteria into the starter culture. Previous studies have demonstrated this to be the strongest influence on the development of the microbial communities [38]. However, not all Zambian samples contained this root material and samples with added root material were not significantly different compared to Zambian samples without added root material (Appendix A).

The result that differences in temperature could be the main force resulting in communities with different bacterial community structure is in line with earlier work that had found that raw materials and associated composition of the microbiota did not determine the final microbial community composition [7,9,39]. This work experimentally analyzed the microbial community composition before and after fermentation in traditional maize based fermented food Munkoyo [7]. Other previous studies showed that environmental selection of different pH did affect microbial community composition—this is work performed with Mabisi [8]. Thus, apparently, the selective forces of the fermentation environment have a great impact. In the design of our study, we chose to focus on the potential effect of the environment by sampling similar foods at climatically different locations. The survey based approach, which we used in the present study, complements the more directed experiments that were reported previously.

The results and analysis that suggest a correlation between temperature and thermophile abundancies should be interpreted with caution. Most genera were associated with mesophilic bacteria, and the vast majority of differentially abundant genera were underrepresented in the samples from Zambia. Therefore, in the list of differentially abundant genera, any genera associated with thermophiles would most likely be found to be underrepresented in Zambia. Furthermore, the thermophilic properties were based on reports for specific species or strains that were part of the genus. It was not clear in the study, which species or strain the ASVs and genera corresponded to. There was no data on pH, humidity, exact preparation steps, time of sampling, or the fermentation duration.

One way to test temperature dependence of the microbial communities in the fermented products, would be to collect a number of samples in the summer and the same amount in the winter in the same place, the same producer and the same production process. Based on the conclusions of this study, the samples taken in the winter should be significantly different from the samples in the summer. In an analogous study, samples with and without roots added could be compared. In this study, no significant difference was found between samples from Zambia with added root material and without. The differential abundance analysis of the ASVs provides the information for the construction of a species co-occurrence network [40]. Such a network would connect ASVs that are found together much more or much less often than would be predicted by chance. Such a network allows the modelling of cooperation and inhibition of bacteria within the communities. The presence of one species of bacteria may be used to predict the presence of associated bacteria in the co-occurrence network. Some genera that were highly abundant in samples from Benin and Tanzania, were completely absent in the samples from Zambia. Genera that were differentially abundant across samples form interesting candidates for species distribution modelling [41]. A species distribution model (otherwise known as an ecological niche model) is a statistical equation based on species observation locations and environmental data. The abundance count values would have to be converted to binary species occurrence data with specified thresholds. Environmental data such as production details and local weather would constitute the environmental data. The resulting models could be used to form predictions of the occurrence of the ASVs in new products, and provide insight into the importance of different environmental factors, similar to linear regression modelling.

While not a specific objective of our work, it is interesting to note that through our culture independent approach of 16S amplicon sequencing, we did not detect any pathogenic bacteria, at least not at an abundance of >0.5%. This is in line with other (published and unpublished) work where similarly no pathogens were found in mature fermented foods at the end of a fermentation cycle [42]. One of the key properties of (traditional) fermentation is the lowering of pH. Prolonged exposure to low pH suppresses most pathogens to low and acceptable levels 2 [43,44]. This does not mean that pathogens are surely completely absent. Specific isolation and selective plating could reveal presence of pathogenic bacteria at low levels.

QIIME2 proved to be an effective tool for the initial data processing in this project. It allowed for reproducible results in data provenance, and the Jupyter Notebook environment allowed for a more presentable workflow compared to the traditional command-line environment. The results from QIIME2 were formatted in universal file types such as biom-files for 12 feature-tables and nwk-files for phylogenetic trees. These advantages allowed for a separation of processes where the initial, computationally expensive steps were performed by QIIME2 on server, and the downstream, computationally inexpensive steps were performed on local PC’s using opensource R-packages. The results confirm the hypothesis that traditionally fermented products from different countries harbor significantly different microbial communities. Samples from Zambia deviate the most from the other samples. This deviation was evident in the lower alpha diversity on average and the higher dispersion of diversity (Figure 4), the tendency for samples to cluster together (Figure 5 and Figure 6), the PERMANOVA’s involving Zambia (Table 3, columns 1, 3 and 4) and the differential abundance analysis (Table 3 and Figure 7).

The use of the 16S rDNA gene restricted microbial diversity so that we could only detect bacteria, mostly, in a genus level. As a consequence, a study of this type leaves out potentially significant players in the microbial community such as yeasts and bacteriophages. Another challenge of the study was the high variability between samples of the same category. Samples within the countries differed in main ingredients and enzyme source, and the names of the producers were not documented. Seven out of the twelve samples from Zambia were produced with root-based enzymes, and two out of the eleven included samples from Benin contained millet instead of maize. Products from Tanzania had all the same ingredients but with different ratios. Such variation allows for additional analysis within individual countries, but also further complicates the interpretation of data that is, even under controlled conditions, stochastic in nature [8,45].

In conclusion, our results show that environmental selection pressure can shape microbial community structure in natural microbial communities derived from traditional cereal-based fermented foods. This highlights that traditional fermentation can be viewed as a natural ecological process, which has implications for the development of starter cultures. For instance, our results suggest that once selection pressures (environmental conditions, substrates, processing conditions) are uniform, this will result in foods with a similar microbial community composition, suggesting similar product properties. The effect of these selection pressures should now be validated in directed sampling or (field) experiments with treatments contrasting fermentation temperature and other environmental parameters while controlling for factors such as processing procedure and raw materials. In this way, further research in this direction to test these ideas would further bridge fundamental insight of ecology to the application of (traditional) fermentation [2].

## Figures and Tables

**Figure 1 microorganisms-10-01354-f001:**
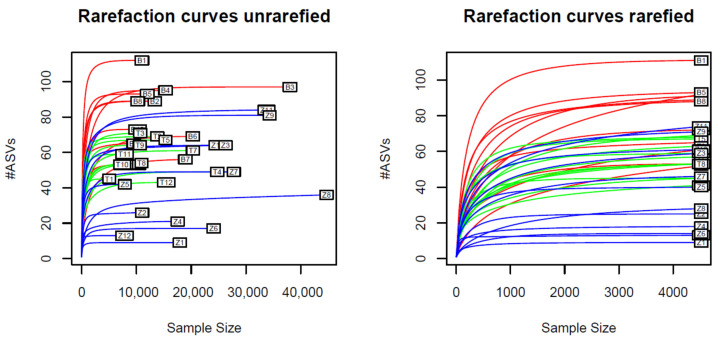
Rarefaction curves before and after sample rarefaction. Every curve corresponds to a single sample. Red curves correspond to Benin samples, green curves correspond to Tanzanian samples and blue curves correspond to Zambian samples. The curves were obtained by plotting the number of unique ASVs corresponding to random subsamples of increasing size (incrementation step size: 50 amplicons). All curves appear to follow asymptotic growth.

**Figure 2 microorganisms-10-01354-f002:**
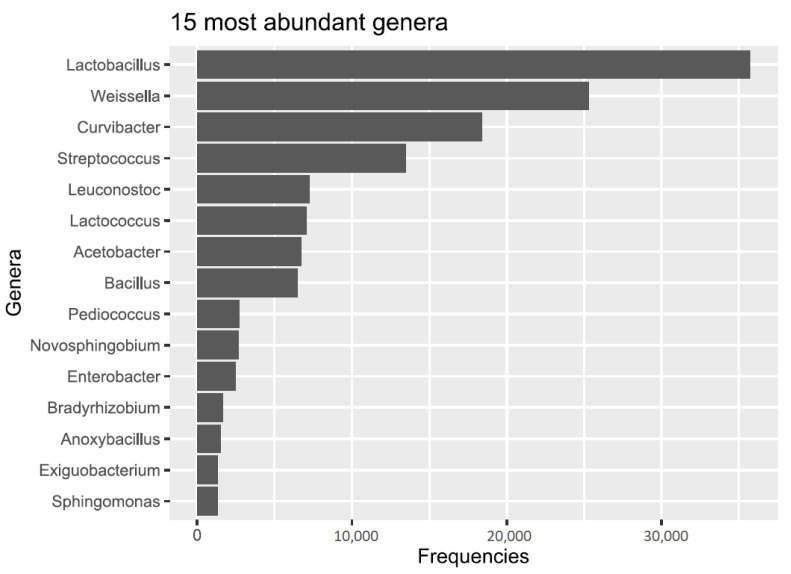
Abundant genera frequencies (counts of the number of sequence reads found). These fifteen genera account for almost 88% of the total bacterial abundance. Key players were ASVs that were classified as lactic acid bacteria such as *Lactobacillus*, *Weisella* and *Streptococcus*, and proteobacteria such as *Curvibacter* and *Acetobacter*.

**Figure 3 microorganisms-10-01354-f003:**
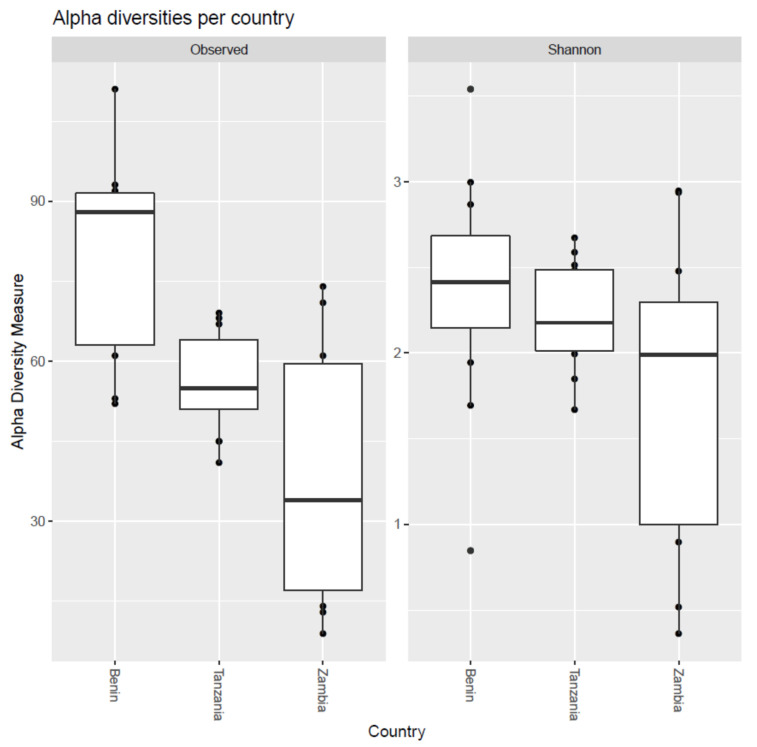
Alpha diversities per country according to species richness (Observed) and Shannon diversity. The boxplots portray the highest diversity in the Beninese samples, and the lowest diversity in the Zambia samples. The species richness was significantly higher for the Benin samples compared to the Tanzanian and Zambian samples, while Tanzania and Zambia were not significantly different in terms of sample richness. The Shannon diversities revealed no significant difference between the countries.

**Figure 4 microorganisms-10-01354-f004:**
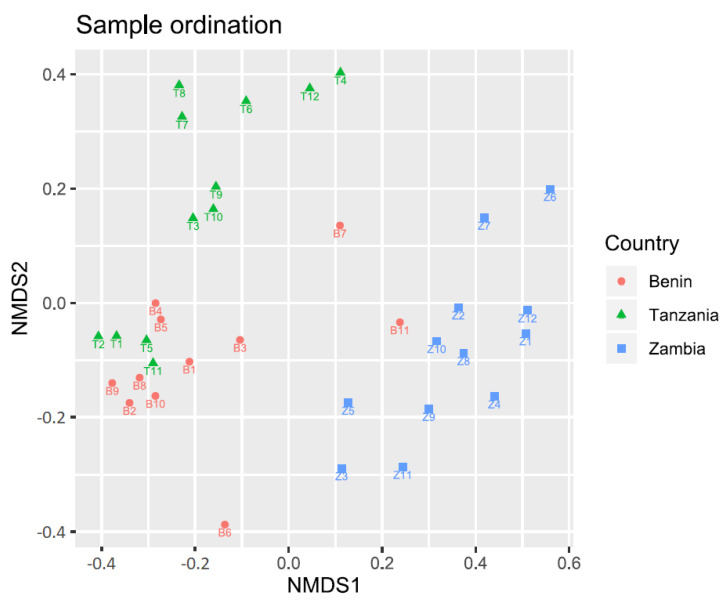
NMDS ordination based on Unifrac sample distances. The samples from Zambia (blue, square) were generally separated from the other samples along the horizontal axis. The samples from Benin (red, circle) and Tanzania (green, triangle) were generally separated along the vertical axis.

**Figure 5 microorganisms-10-01354-f005:**
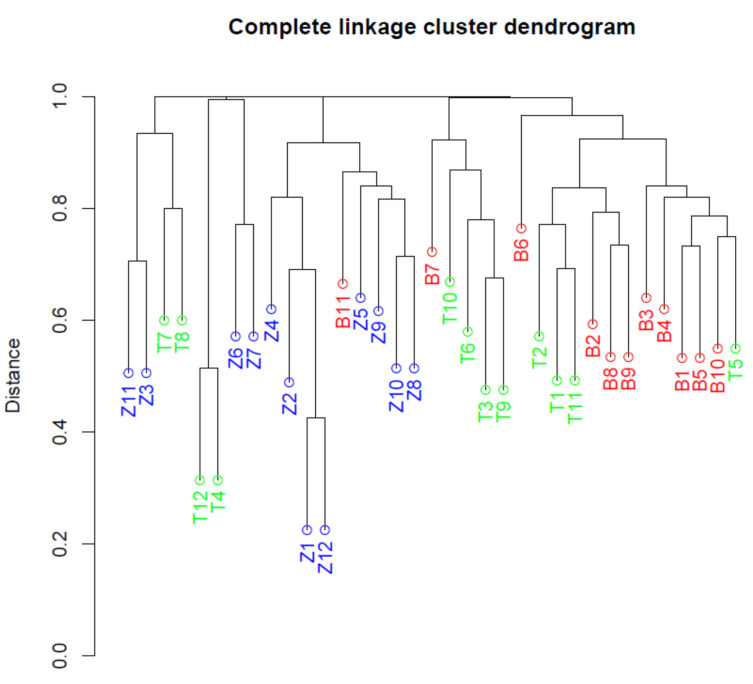
Clustering dendrogram based on Unifrac sample distances. Here, the branches are labelled by sample ID, of which the capital letter corresponds to the first letter of the country. The dendrogram revealed that samples from the same country tended to cluster together, and that the samples from Zambia were much more dispersed than the other samples.

**Figure 6 microorganisms-10-01354-f006:**
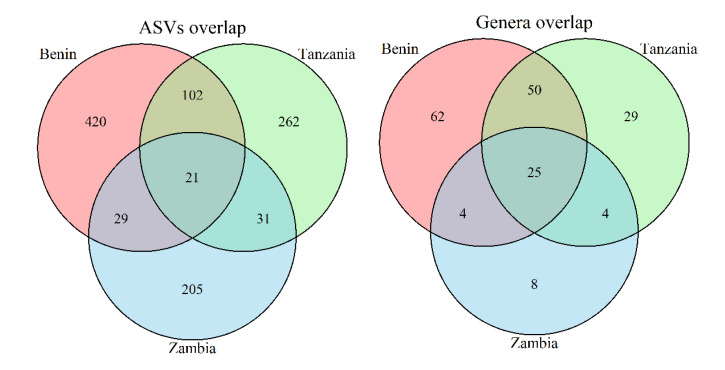
Overlap of ASVs and genera between samples from the three countries. The high number of unique ASVs corresponds to much lower numbers of corresponding genera (non-overlapping areas in the diagrams), whereas the number of shared ASVs is only slightly higher than the number of shared genera.

**Figure 7 microorganisms-10-01354-f007:**
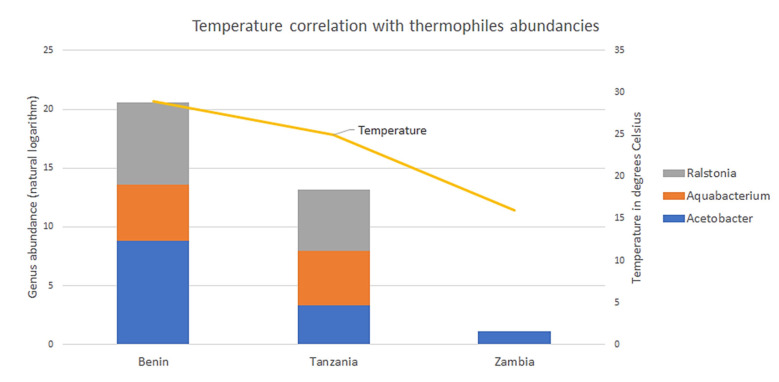
Temperature and thermophiles abundancies. The stacked bar charts represent the abundance of the three thermophilic and thermotolerant genera (y-axis on the left). The line represents the temperature of the locations of the samples (y-axis on the right). All three thermophiles appeared to favor the higher temperature samples from Benin, and to a lesser extent Tanzania.

**Table 1 microorganisms-10-01354-t001:** ASV distribution. All 597,755 amplicon sequences were assigned to 1127 different ASVs. The distribution of observations per ASV was highly skewed as the mean value was more than fifteen times as high as the median value.

Minimum	First Quartile	Median	Third Quartile	Mean	Maximum
1	12	34	113	~530	73,787

**Table 2 microorganisms-10-01354-t002:** Amplicons and ASVs before and after rarefaction. Rarefaction through sub-sampling resulted in an equal number of amplicon observations in each sample. This reduced the total number of amplicons considerably and resulted in the removal of 57 low-abundance ASVs.

	# Amplicons Total	# Amplicons in Samples	Number of ASVs
Before rarefaction	597,755	5021–44,710	1127
After rarefaction	158,130	4518	1070

**Table 3 microorganisms-10-01354-t003:** Genera distribution. All 182 genera had at least one ASV, and at least one corresponding amplicon assigned to them. Although half of the genera were assigned to 20 amplicons or less, the much higher mean value of ~838 observations per genus reveals that a small number of genera account for most of the observations. The most abundant genus (‘Maximum’ frequency) corresponded to nearly one-fourth of all the observations.

Minimum	First Quantile	Median	Third Median	Mean	Maximum
1	4	20	157	~838	35,696

**Table 4 microorganisms-10-01354-t004:** PERMANOVA results summary. Orange (darker) cells correspond to low values and yellow (lighter) cells correspond to high values relative to other values in the respective row. All models that incorporated the samples from Zambia were characterized as significant between-group variation. The only PERMANOVA that yielded no significance was Benin vs. Tanzania.

	All Countries	Benin vs. Tanzania	Benin vs. Zambia	Tanzania vs. Zambia
Between group variation	0.185	0.072	0.179	0.183
Within group variation	0.815	0.928	0.821	0.817
Total variation	1	1	1	1
F value	3.628	1.634	4.582	4.933
Pr (>F)	0.001	0.015	0.001	0.001

**Table 5 microorganisms-10-01354-t005:** Number of overabundant ASVs in country-wise comparisons. Each row displays numbers of differentially abundant ASVs in a comparison. The country comparison is defined in the first two columns. Column ‘A’ represents the numbers of ASVs that were overrepresented in country A, compared to country B. Column ‘B’ represents the opposite. The second column from the right represents the total numbers of differentially abundant ASVs, and the right-most column displays the total number of ASVs of the two countries, shared or otherwise.

Country A	Country B	Overabundant ASVs in A	Overabundant ASVs in B	Total Differentially Abundant ASVs	Total Number of ASVs
Benin	Tanzania	30	22	52	858
Benin	Zambia	60	31	91	803
Tanzania	Zambia	53	27	80	654

## Data Availability

All data associated with this work will be made publicly available upon acceptance of the manuscript. Raw sequencing reads have been deposited to NCBI SRA database under the project number PRJNA854902 (https://www.ncbi.nlm.nih.gov/sra/PRJNA854902 accessed on 4 July 2022).

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
