# Peer review of "Environmental Selection Shapes Bacterial Community Composition in Traditionally Fermented Maize-Based Foods from Benin, Tanzania and Zambia"

_microorganisms, 2022, doi:10.3390/microorganisms10071354_

Round 1

Reviewer 1 Report

The Mauscript focuses on the identification of microbes in fermented foods from local markets in Benin, Tanzania and Zambia. A bioinformatic analysis was performed using the QIIME2 bioinformatics platform

Detailed comments:

- The purpose of the research was not clearly defined;

- What does "traditional fermentation" mean? The samples came from production plants or from a kitchen (home) factory?

-The chemical composition of the tested samples has not been determined, in particular the carbohydrate content (including potential carbon sources), which has a significant impact on the growth and survival of microorganisms. The authors should pay more attention to the composition of the raw materials from which fermented foods are made.

- When identifying the number and type of microorganisms, the authors should complete the analysis and conclusions with information on pathogenic microorganisms. Can the labeled pathogens cause gastroenteritis after eating such food? Which pathogenic microorganisms are there the most?

Author Response

*** We wish to thank the Reviewer for the very useful comments on our manuscript. We have made every effort to fully address all comments as we describe below.

- The purpose of the research was not clearly defined;

*** We now realize that we did not provide sufficient context to our work – Reviewer one gave a similar comment. Previous work has suggested that the environment is important in shaping the microbial community structure in microbial systems such as the microbial communities in traditional fermentation. Previous work using directed experiments has shown that a variation in raw materials used can result in similar microbial communities and that on the contrary variations in fermentation temperature do affect microbial composition. We thus here sought out to use a survey approach to ask if climatic conditions can affect microbial composition in traditional foods that use similar processes and raw materials. We have now made this more clear by adding text to the introduction and the discussion section. See lines 65-70 and 477-485.

- What does "traditional fermentation" mean? The samples came from production plants or from a kitchen (home) factory?

*** Traditional means that no formal process was used, so the products were made at home by local processors. We have clarified this in the methods section, see line 99.

-The chemical composition of the tested samples has not been determined, in particular the carbohydrate content (including potential carbon sources), which has a significant impact on the growth and survival of microorganisms. The authors should pay more attention to the composition of the raw materials from which fermented foods are made.

*** We agree with the Reviewer that this is an important point. We value the comment that properties of the raw material and properties of the final product could have value to explain variation in microbial community structure. However, perhaps surprisingly, in some of our earlier work we had found that raw materials and associated composition of the microflora did not determine the final microbial community composition. This was work by Phiri et al who systematically analyzed the microbial community composition before and after fermentation took place using the traditional maize based fermented food Munkoyo. Other previous studies showed that environmental selection of different pH did affect microbial community composition – this is work performed with Mabisi. Thus, apparently, the selective forces of the fermentation environment have a great impact. In our design we had chosen to focus on potential effect of environment by sampling similar foods at climatically different locations. The survey based approach we followed here complements the more directed experiments that were reported previously.

We have modified our manuscript by making the above reasoning more clear. We have added three sentences to the introduction and a paragraph to the discussion section. See lines 64-71 and 478-489. 

- When identifying the number and type of microorganisms, the authors should complete the analysis and conclusions with information on pathogenic microorganisms. Can the labelled pathogens cause gastroenteritis after eating such food? Which pathogenic microorganisms are there the most?

*** While not a specific objective of our work, it is indeed interesting to specifically check for the presence of pathogenic bacteria since we are using a traditional food product. Using our culture independent approach of 16S amplicon sequencing, we did not detect any pathogenic bacteria, at least not at an abundance of >0,5%. This is in line with other (published and unpublished) work where also no pathogens were found in mature fermented foods at the end of a fermentation cycle. One of the key properties of (traditional) fermentation is the lowering of pH. Prolonged exposure to low pH suppresses most pathogens to low and acceptable levels. This does not mean that pathogens are surely completely absent. Specific isolation and selective plating could reveal presence of pathogenic bacteria at low levels. We have added a discussion on this issue in the Discussion section of our paper. See line 523-530.

Reviewer 2 Report

Interesting work. However, the physical-chemical analysis of the final product should be analyzed, since it has a great influence on the modulation of the microbial community that can be established as well as the initial microflora present in the raw material.

Author Response

*** We thank the reviewer for this very positive review and the useful input. The other Reviewer made a similar comment. We value the comment that properties of the raw material and properties of the final product could have value to explain variation in microbial community structure. However, perhaps surprisingly, in some of our earlier work we had found that raw materials and associated composition of the microflora did not determine the final microbial community composition. This was work by Phiri et al who systematically analyzed the microbial community composition before and after fermentation took place using the traditional maize based fermented food Munkoyo. Other previous studies showed that environmental selection of different pH did affect microbial community composition – this is work performed with Mabisi. Thus, apparently, the selective forces of the fermentation environment have a great impact. In our design we had chosen to focus on potential effect of environment by sampling similar foods at climatically different locations. The survey based approach we followed here complements the more directed experiments that were reported previously.

We have modified our manuscript by making the above reasoning more clear. We have added three sentences to the introduction and a paragraph to the discussion section. See lines 63-70 and 477-485.

Round 2

Reviewer 1 Report

Dear Authors,

You took all my comments into account when revising the manuscript. I have no more comments. I recommend it for publication.

Author Response

We thank the reviewer for this very positive and encouraging reply.